



# Investigating coastal backwater effects and flooding in the coastal zone using a global river transport model on an unstructured mesh

Dongyu Feng[1], Zeli Tan[1], Darren Engwirda[2], Chang Liao[1], Donghui Xu[1], Gautam Bisht[1], Tian Zhou[1], Hong-Yi Li[3], L. Ruby Leung[1]

[1]Pacific Northwest National Laboratory, Richland, WA, 99354, USA
[2]T-3 Fluid Dynamics and Solid Mechanics Group, Los Alamos National Laboratory, Los Alamos, NM, 87545, USA
[3]Department of Civil and Environmental Engineering, University of Houston, TX, 77204, US

*Correspondence to*: Zeli Tan (zeli.tan@pnnl.gov)

**Abstract.** Coastal backwater effects are caused by the downstream water level increase as the result of elevated sea level, high river discharge and their compounding influence. Such effects have crucial impacts on floods in densely populated regions but have not been well represented in large-scale river models used in Earth System Models (ESMs), partly due to model mesh deficiency and oversimplifications of river hydrodynamics. Using two mid-Atlantic river basins as a testbed, we perform the first attempt to simulate the backwater effects comprehensively over a coastal region using the MOSART river

transport model under an Earth system model framework i.e., Energy Exascale Earth System Model (E3SM) configured on a regionally-refined unstructured mesh, with a focus on understanding the backwater drivers and their long-term variations. By including sea level variations at the river downstream boundary, the model performance in capturing backwaters is greatly improved. We also propose a new flood event selection scheme to facilitate the decomposition of backwater drivers into different components. Our results show that while storm surge is a key driver, the influence of extreme discharge cannot be

neglected, particularly when the river drains to a narrow river-like estuary. Compound flooding, while not necessarily increasing the flood peaks, exacerbates the flood risk by extending the duration of multiple coastal and fluvial processes. Furthermore, our simulations and analysis highlight the increasing strength of backwater effects due to sea level rise and more frequent storm surge during 1990-2019. Thus, backwaters need to be properly represented in ESMs for improving predictive understanding of coastal flooding.

Keywords: Earth System Model, hydrologic modeling, backwater effects, compound flood, unstructured mesh, sea level rise



## 1 Introduction

Backwater zones are regions at the river downstream sections, a fluvial-marine transition area between upstream flow and
estuary/coastal river plume, where the river flow is affected by the coastal processes, such as sea level changes, tides and
storm surge (Lamb et al., 2012), and can extend hundreds of kilometers upstream in low-lying watersheds (e.g., up to 500
km in Mississippi River). Coastal backwaters are created by elevated sea level that can cause upstream propagation of flood
waves and the attenuation of the spatial and temporal water stage fluctuations (Luo et al., 2017). These effects play a critical
role in floodplain storage and river discharge (Paiva et al., 2013) and also have a key influence on the biogeochemistry and
geomorphology at the terrestrial-aquatic interface (Dykstra & Dzwonkowski, 2020; Lamb et al., 2012; Ward et al., 2020).
With population growth near coastal regions (Tellman et al., 2021), coastal backwaters are expected to exert a greater impact
on human and natural systems.

The backwater zones usually face severe flood risks as a result of tide, storm surge, rainfall runoff and their combined
effects. During a landfalling hurricane with strong winds and heavy rainfall, storm surge drives coastal waters to propagate
into the river network and interact with high river discharge (Bilskie & Hagen, 2018). When multiple drivers occur
simultaneously or in close successions, the flood event is referred as compound flooding (Santiago-Collazo et al., 2019).
Coastal backwater induced floods have strong temporal and spatial variabilities (Hendry et al., 2019), depending largely on
the local topography and storm characteristics (Gori et al., 2020). Due to climate warming, the frequency and intensity of
such compound flooding have exhibited an increasing trend (Bates et al., 2021; Rahmstorf, 2017), as a result of intensified
storm surge (Camelo et al., 2020; Marsooli et al., 2019), more frequent extreme precipitation (Alfieri et al., 2016) and
accelerated sea level rise (SLR) (Kulp & Strauss, 2019; Orton et al., 2019). Although SLR and storm intensification are
considered the most influential flooding drivers (Hwang et al., 2020), projected changes in river discharge also play an
important role in modulating the flood potentials (Bermúdez et al., 2021).

Understanding the backwater drivers is prerequisite to mitigating the related flood risks. However, the interactions among
the backwater drivers and their respective contributions through fluvial processes, storm and climate are not well understood
(Dykstra & Dzwonkowski, 2021). Streamflow in the backwater zones is affected by river topology, upstream discharge, sea
level variations and their interactive effects (Castelltort et al., 2020; Hellmers & Fröhle, 2022). Specifically, river topology is
characterized by the river channel geometry, riverbed elevation and river's receiving water body. Among these factors,
riverbed elevation, due to its control on the backwater propagation extent, has been widely recognized in previous studies
(Gori et al., 2020). In contrast, the river's receiving water body has not yet drawn much attention. As rivers contribute to a
variety of water bodies including deltaic floodplains, estuaries and coastal oceans (Mikhailov & Gorin, 2012), the interactive
effects of river discharge and sea level vary substantially. For example, the impact of river discharge on the local sea level is
much more intense in a narrow tidal-river estuary than in an open sea (Rayson et al., 2015; Chegini et al., 2022). In a narrow
estuary, flood risks are further exacerbated because high discharge increases the local sea level, high sea level induced by





storm surge impedes river discharge to the ocean, and the interaction of these two mechanisms intensifies the backwater effects (Eilander et al., 2020).

    Large-scale river models are one of the major components of Earth System Models (ESMs) that couple the atmosphere, land, river, and ocean models to simulate the global water cycle (e.g., Golaz et al., 2019; Leung et al., 2020) and assess flood risks (Hirabayashi et al., 2013; Towner et al., 2019). Although hydraulic or hydrodynamic models were used more often in

previous studies to simulate storm surge induced coastal inundation (Bakhtyar et al., 2020; Muñoz et al., 2020), there have been growing applications of large-scale river models to assess the compound fluvial and coastal flooding at basin (Chen et al., 2013), regional (Ikeuchi et al., 2017; Yamazaki et al., 2012) and global scales (Eilander et al., 2020; Mao et al., 2019) because they are more computationally efficient and can be coupled directly to other components of the earth system. However, several limitations in the current generation of ESMs impair the realistic representation of coastal backwaters and

human–land–river–ocean interactions at the terrestrial-aquatic interface (Ward et al., 2020). First, most ESMs are configured with one-way coupled river and ocean models, in which water only flows from rivers to oceans and the impact of elevated sea levels on upstream river stage is ignored. Second, the meshes used in most ESMs are too coarse to represent backwater effects. For example, in the high-resolution configuration of Energy Exascale Earth System Model (E3SM), a uniform resolution of 12.5 km is used for the river model (Caldwell et al., 2019). The resolutions of other widely-used large-scale

river models also only range from 5 km to 25 km. Much higher spatial resolutions (~ km) are required to resolve the smaller-scale topology near the coastline (Bates et al., 2021; Trigg et al., 2016) for coastal backwaters. Last, while most existing ESMs apply structured meshes in their river components, unstructured meshes are needed to achieve more flexible variable resolutions within areas of interest, such as high resolutions along the coastline, as well as to accommodate the high spatial variation of coastal processes.

Motivated by the increasing flood risks in a warming climate, this study is part of a larger effort to develop capabilities in representing land-river-ocean interactions in E3SM for modeling the changing compound flood risks in coastal regions and the potential implications for the regional and global water and biogeochemical cycles. More specifically, the objectives of this study are to (a) assess the capability of two-way coupled river and ocean models on a regionally-refined mesh to capture coastal backwater effects; and (b) understand the major and interactive backwater drivers and their long-term variations

under climate change in two contrasting coastal river basins. The backwater drivers are decomposed using a novel extreme flood event selection scheme. Each selected event is identified by the dominant flood drivers. In Section 2, we provide an overview of the study domain, the river model, the unstructured mesh, and the methods of extreme event selection and drivers decomposition. Model evaluation and analyses are provided in Sections 3, 4 and 5. In Section 6, we discuss the findings and limitations. Finally, the conclusions are provided in Section 7.



## 2 Methodology

This section describes the study domain of two mid-Atlantic river basins. A global river routing model on a regionally-refined unstructured mesh and with two-way river-ocean coupling physics is introduced. We also develop a method to select extreme events and decompose the flood drivers of the selected events.

### 2.1 Study domain

The mid-Atlantic region of the US is exposed to frequent tropical cyclones that bring intense precipitation and storm surge (Sun et al., 2021). In this study, we define the mid-Atlantic region as Susquehanna River Basin (SRB) and Delaware River Basin (DRB) (Fig. 1). Susquehanna River drains 71,228 km$^2$ to the northern end of Chesapeake Bay, contributing ~50% of freshwater inflow to the estuary (Leathers et al., 2008). Chesapeake Bay is the largest estuary in the US with a surface area of 11,601 km$^2$ and a shoreline extending over 7000 km. Chesapeake Bay has varied tidal characteristics across the estuary, e.g., mixed tide in the northern portion and semidiurnal tide near the bay mouth. In addition to Susquehanna River, several other large rivers also drain to this estuary. For Chesapeake Bay, the amount of the freshwater outflow is approximately the same as the seawater inflow from the mid-Atlantic coastal waters (Valle-Levinson, 1995). Delaware River drains 35,070 km$^2$ to Delaware Bay and contributes 58% of freshwater to the estuary (Whitney & Garvine, 2006). Delaware Bay has 2,030 km$^2$ in surface area and is dominated by semidiurnal tide. The tidal range is 1.5 m at the bay mouth and increases towards Trenton (USGS gauge 01463500 in Figure 1). Trenton is referred as the downstream limit of freshwater (Sharp, 1983), as there is a hydraulic jump at 2.7 km downstream of the station due to an abrupt decrease in channel bathymetry (Zhang et al., 2020). Together, SRB and DRB have over 4 million residents and DRB provides drinking water to 6% of the US population. The locations of in-situ observations are provided in Figure 1. Among over 100 USGS gauges in the mid-Atlantic region, we selected all USGS gauges in the mainstem of Susquehanna River and Delaware River, respectively, for simulated streamflow validation. The water level data at 6 NOAA tidal gauges are also selected for data analysis and/or model validation. While the coastal tidal gauge (8534720) is used for identifying storm surge, the two tidal gauges near the river mouths (8573364 and 8545240) provide the downstream boundary condition (BC) for the river model. The four tidal gauges at the downstream section of Delaware River (8545240, 8546252, 8539094 and 8548989) are used in model evaluation.

### 2.2 Global River Routing Model

The Model for Scale Adaptive River Transport (MOSART) (Li et al., 2013; Li et al., 2015), the river component in E3SM (Golaz et al., 2019) is used for river modeling. MOSART is a river routing model applicable across local, regional, and global scales. The model is driven by runoff from a land surface model and simulates water flow from hillslopes to tributary subnetworks and to main channels. The routing schemes in MOSART include kinematic wave and diffusion wave equations, two simplified forms of the 1-dimensional Saint Venant equations. The routing of surface runoff in hillslopes and tributary is





represented using the kinematic wave method. The flow in the main channel is represented by the diffusive wave method. The momentum equation in the diffusive wave method is (Chow, 1988)

$$\frac{\partial h}{\partial x} - S_0 + S_f = 0, \tag{1}$$

where $h$ is the water depth in the channel, $S_0$ is riverbed slope and $S_f$ is the friction slope. Compared to the diffusive wave method, the kinematic wave method neglects the first term of Eq. 1 in its momentum equation. The flow velocity ($v$) is

estimated using the Chezy-Manning equation

$$v = \frac{|S_f|}{S_f} n^{-1} R^{\frac{2}{3}} |S_f|^{\frac{1}{2}}, \tag{2}$$

in which Manning's $n$ is used as the frictional coefficient and $R$ is the hydraulic radius. The backwater effect can be represented in the diffusive wave method, as the flow velocity is determined by both the riverbed slope ($S_0$) and the water level variations ($h$) along the river channels (Luo et al., 2017). In extreme conditions, when the downstream water stage is

higher than that of the current channel, $S_f$ becomes negative, resulting in a backwater. This phenomenon is recently observed in Mississippi River during Hurricane Ida (Miller, 2021).

The cross-section of the main channel is specified as rectangular in MOSART when channel water depth is no more than the bankfull depth ($H$). The channel width ($W$) and bankfull depth ($H$) are estimated from the total upstream drainage area ($A_{total}$) using empirical formulations (Bent & Waite, 2013):

$$W = a(A_{total})^b, \tag{3}$$

$$H = a(A_{total})^b, \tag{4}$$

where $a$ and $b$ are empirical parameters. When channel water depth exceeds $H$, an elevation profile is invoked to capture the elevation variation in the floodplain (Luo et al., 2017).

In this study, the runoff inputs for MOSART are obtained from Global Reach-level Flood Reanalysis (GRFR) (Yang et al.,

2021), an offline simulation from a high-resolution land surface model that has been calibrated and bias corrected. The original configuration of the MOSART diffusive wave method applies a static coastal boundary condition (CBC), i.e., either normal depth or fixed mean sea level at the river mouth (Luo et al., 2017). The normal depth boundary assumes that the friction slope ($S_f$) equals to the riverbed slope ($S_0$) at the river outlet cell. This simplification, while reasonable for global simulations in which the influence of coastal processes is limited, can be problematic in low-lying coastal regions. To

represent the backwater effects induced by the dynamic sea level variation, this study introduces a new, dynamic CBC option in MOSART to read in time-varying water level data at a time interval consistent with the land-river coupling frequency in E3SM. The coupling time interval is set as 1 hour in this study. This dynamic CBC is only configured for the rivers of interest, while the static CBC is used in all other outlet boundaries due to the limited data availability.



**2.3 Coastal refined global unstructured mesh and flow direction map**

A global unstructured mesh with a resolution of ~100 km has been developed using the JIGSAW mesh library (Engwirda & Ivers, 2016; Engwirda, 2017), which enables: (a) the flexibility of embedding high-resolution subdomain within the ESM's global configuration; (b) oversampled geometrical features (O(<100m)), e.g., river network and coastline, to be simplified to coarser ESM length scales on orders of 2-60km; (c) close alignment of the complex geography of coastline, watershed boundaries and river networks (Engwirda & Chang, 2021). The global mesh is developed to allow for more seamless

coupling of the land, river and ocean components in E3SM (Fig. 2) for more consistent modeling of global surface processes. In the mid-Atlantic region, the mesh resolution is refined to ~3 km to better resolve local coastal and watershed processes (Fig. 2b). Significant effort is also made to ensure that the cells in the high-resolution mesh match the prescribed dam locations and the orientations of the edges conforming to the flow direction along the main channel.

    The river networks and flow directions are modeled using HexWatershed (Liao et al., 2020; Liao et al., 2022; Liao, 2022), a

watershed and flow direction model that supports both structured and unstructured meshes for river routing models. HexWatershed uses a topological relationship-based approach to define river networks in the mid-Atlantic region (Lehner et al., 2008). To generate the flow direction for the entire domain, HexWatershed uses a hybrid depression filling and breaching stream burning algorithm to remove local depressions while minimizing modifications to surface elevation and produces flow routing parameters including the flow direction map, channel slope, and drainage area, which are critical for accurately

representing coupled land-river-ocean processes.

**2.4 Extreme event selection**

    The extreme events of fluvial flood (FF) and storm surge (SS) are separately selected based on their corresponding time-series observations. The selection of FF follows the strategy proposed in (Zhang et al., 2021). In the time-series of discharge data, a flood event is identified by first selecting the flood peaks using the peaks-over-threshold (POT) approach (Lang et al.,

1999). The threshold is determined based on automatic threshold selection and the independence of the peak series is examined with a declustering method (Zhang et al., 2021). The start and end dates are specified using the empirical formulation:

$$Q_S \leq aQ_P; \; T_P - T_S \leq b(5 + \ln(\frac{A}{1.609^2})), \tag{5}$$

$$Q_E \leq aQ_P; \; T_E - T_S \leq b(5 + \ln(\frac{A}{1.609^2})), \tag{6}$$

where $T_P$, $T_S$ and $T_E$ are the peak date, start date and end date, $Q_P$, $Q_S$ and $Q_E$ are the discharge on the corresponding date, and $A$ is the basin drainage area. The empirical parameters $a$ and $b$ are specified as 0.5 and 1.5, respectively.

    The selection of storm surge (SS) is performed in three major steps. The 1st major step is to extract the SS component from the hourly total water level (TWL) data at the NOAA tidal gauge (i.e. 8534720 in Figure 1): (i) the TWL time series is detrended by extracting the annual mean sea level, which removes the effects of SLR; (ii) the predicted astronomical tides



are then derived from the harmonic tidal analysis performed on the detrended TWL on a year-by-year basis using 8 major tidal constituents; (iii) SS is the non-tidal residual obtained by extracting the tides from the detrended TWL.

The 2$^{nd}$ major step is to filter the extreme SS events using a peak detection algorithm (Brakel, 2014). When a data point is $m$-fold standard deviations away from the moving mean, an event peak is identified. The start and end times of the corresponding event are the nearest data points that change signs. In addition to $m$, the other input parameters of this

algorithm are *lag* and *influence*: *lag* is the number of observations to smooth the data, or the length of the moving window; *influence* represents the influence of new signals on the threshold. Here we performed the SS event selection by setting $m = 5$, *lag* $= 30$ and *influence* $= 0$. These parameters are determined to ensure that the selected SS events include all documented hurricane events in the mid-Atlantic region.

The 3$^{rd}$ major step is to select the extreme SS events of interest by extracting the events with SS peaks larger than the 99.5th

percentile. When there is an overlap between the FF and SS events, a compound flood event is identified, for which the duration is defined as the combined period of the two events. The applied event selection method can be more accurate than those used in continental and global applications, where the event period is simply determined using a predefined time-window (e.g. ±1 days of a peak event) (Nasr et al., 2021; Ward et al., 2018; Wu et al., 2021).

## 2.5 Decomposition of backwater drivers

The backwater drivers are decomposed into three different levels (Fig. 3). The first level considers the river topology and the forcing that affects the river flow directly. The direct forcing in the backwater zone is considered as the upstream discharge and the TWL at the river mouth. To address the impact of topology, we compare the backwater effects in Susquehanna River and Delaware River, which differ significantly in terms of the riverbed elevation along the downstream section and the receiving water body. As previous studies have shown the crucial impact of sea level variations on the backwater effects

(Yamazaki et al., 2012), we further decompose the TWL into low-frequency surge (LFS) and tide in the second level. The predicted tide is estimated using harmonic tidal analysis and the LFS is obtained by subtracting the tide from the TWL.

The third level decomposes the LFS to high discharge, SS and their compound effects. It is worth noting that the LFS differs from the SS in that the LFS is extracted from the TWL at the NOAA gauge nearest to the river mouth (e.g., 8545240 in Figure 1) where the river discharge can have a significant influence on the LFS. In particular, the LFS is dominated by both

river discharge and storm surge in narrow tidal rivers, such as Delaware Bay. In contrast, the river discharge influence is negligible in larger estuaries or coastal oceans, such as Chesapeake Bay. Previous studies mainly focused on the drivers of river discharge and SS (Nasr et al., 2021; Ward et al., 2018), assuming they are distinct mechanisms that have little mutual interactions. Such assumption is valid when the river drains directly to a large receiving water body. However, cases with a river ending within a small estuary have not been explored. In such cases, the high discharge during fluvial flood increases

the water level within the estuary and attenuates the spatial variation of water stage along the river (Luo et al., 2017), creating backwater effects. Thus, in the third level, we attempt to understand the respective role of high discharge, SS and their interactive impacts on the LFS induced backwater effects. These drivers are separated based on the selected flood



events in Section 2.4. Events dominated by the drivers of high discharge, SS and their compound effects are denoted as $LFS \cap FF \cap \overline{SS}$, $LFS \cap SS \cap \overline{FF}$ and $LFS \cap compound$, respectively. The symbol $\cap$ means the intersection period between two events and the overline means to exclude the different drivers if there is an intersection. Each event is measured in terms of the event duration and the peak water level during the event.

The trend analysis was performed to assess the annual trends of the backwater drivers. For each event type, we calculated the annual duration, occurrence and the maximum peak value on an annual basis. The impact of SLR is addressed by selecting the SS event without detrending the TWL data, while the other steps are the same as those in Section 2.4. The corresponding case is referred as SS+SLR. The nonparametric Mann-Kendall (MK) test (Tosunoglu & Kisi, 2017) was employed to statistically assess if there is a monotonic trend. The null hypothesis ($H_0$) and the alternative ($H_a$) are: no monotonic trend is present; and monotonic trend is present, respectively. For each MK test, we set the significance level at 0.1 and calculated the standard MK statistics ($Z$) and $p$-value. The positive (negative) value of $Z$ corresponds to the increase (decrease) trend.

**2.6 Numerical experiments**

MOSART simulations were performed from 1990 to 2019 with the first year excluded from analysis as the spin-up time. This period has sufficient data coverage of both runoff and the water level at NOAA gauges. We configured five simulations based on the aforementioned downstream CBCs: (a) normal depth; (b) total water level (TWL); (c) mean sea level (MSL); (d) low-frequency surge (LFS); (e) tide. The backwater effects are quantified by comparing the TWL and MSL simulations in terms of two quantification metrics: water depth change $\Delta h$ and water volume change $\Delta V$:

$$\Delta h(t,i) = h_{exp}(t,i) - h_{MSL}(t,i), \tag{7}$$

$$\Delta V(t) = \sum_i^N \left( h_{exp}(t,i) - h_{MSL}(t,i) \right) L(i)W(i), \tag{8}$$

where the subscript $exp$ represents TWL, LFS or tide, $h$ is the main channel water depth, $t$ is the model output time step and $i$ is the grid cell index, $L$ and $W$ are the length and width of the main channel within the $i$th cell, respectively, and $N$ is the number of cells with nonzero $\Delta h$.

**2.7 Extreme Events**

The flood events in Delaware River during Hurricane Irene (followed by Tropical Storm Lee) and Hurricane Sandy are used to demonstrate the selection of extreme events and the quantification of the backwater effects. Hurricane Irene, one of the most destructive tropical cyclones in the US history, made its first landfall on the coast of North Carolina on August 27, 2011 as a Category 1 hurricane, followed by another landfall in the southeastern New Jersey on August 27 and a third landfall in New York City. The storm surge peaked at 1.8 m along the coast of New Jersey and the wind speed was up to 105 km/h. The maximum rainfall is 10 inches in DRB. Tropical Storm Lee is the subsequent storm event that formed over the gulf coast and swept the east coast. Lee brought 10~12 inches of precipitation to the mid-Atlantic region, resulting in mainly fluvial processes rather than coastal surges (Ye et al., 2020). The combined events caused two consecutive flow peaks in



Delaware River from Aug 28 to Sep 10, 2011 (Fig. 4). Irene is affected by the interactive storm surge and precipitation-induced fluvial flood and has been studied extensively as an example of compound flood (Xiao et al., 2021; Zhang et al., 2020). Hurricane Sandy, the largest Atlantic hurricane on the US record, made landfall on the coast of New Jersey on October 29, 2012 with a sustained wind speed of 130 km/h. Sandy caused a maximum storm surge of 4 m near New York city and 1.5 m near the Delaware coast. The observed peak flow during Sandy is ∼800 m³/s at Trenton.

## 3 Model evaluation

In this section, the MOSART performance for simulating river discharge and water level is compared between the experiments using the downstream CBC of normal depth, TWL and MSL to demonstrate the importance of imposing an appropriate downstream CBC. The model performance in reproducing the observed water level in the downstream section of Delaware River is significantly improved when the TWL is enforced at the boundary.

### 3.1 River discharge

The MOSART simulated daily discharge is compared with the USGS observations over the simulation period (1991–2019) at the gauges along the main stem of Susquehanna River and Delaware River (Fig. 4). The coefficient of determination $r^2$ and the Kling–Gupta efficiency (KGE) are calculated for each gauge. The MOSART simulation compares reasonably well with $r^2$ and KGE with both over 0.5 across all gauges (Towner et al., 2019). The model performance is generally higher in Susquehanna River and decreases towards the upstream regions. The highest $r^2$ and KGE ($\geq 0.75$) are found at the downstream gauges of the two rivers, as the forcing may not capture the runoff accurately for smaller drainage areas. A closer look at the time series of discharge at these two gauges from 2011 to 2012 shows that the model can capture the hydrograph and smaller peaks of river discharge well, and there is no significant difference of model performance among the different CBC configurations because even the USGS gauges closest to the river outlets are still too far upstream to capture effects of dynamic CBCs. The model has a large bias in Susquehanna River during Hurricane Irene (August 21 to 30, 2011) with the extreme flood peak significantly underestimated. While the observed peak flow is about 20000 m³/s, the simulated flow is about 5000 m³/s. This bias is likely caused by uncertainty in the GRFR runoff forcing (Yang et al., 2021). It is also possible that the USGS discharge that is estimated from a rating curve based on empirical formulations could be biased during extreme events (Di Baldassarre et al., 2012). Overall, the evaluation indicates that the MOSART simulated river discharge reasonably captures the spatial and temporal variability of the observed discharge in the mid-Atlantic region.

### 3.2 Water level

The simulated water level is compared with the observations at four NOAA tidal gauges along the downstream section of Delaware River. The model performance is quantified in terms of $r^2$ and root mean square error (RMSE) (Fig. 5). The TWL simulation results in the best performance, in which $r^2$ is over 0.5 among all the gauges, much higher than the other two



configurations. The lowest RMSE is also obtained from the TWL simulation. By setting the TWL as the downstream CBC,
the model's capability at reproducing the water level variation is greatly enhanced. The same conclusion can also be drawn
in the time-series comparison from 2011 to 2012 (Fig. 6). The TWL simulation accurately captures the small variations in
the observed water level, which are missing in the simulations with normal depth and MSL boundary conditions. The
extreme peaks are overestimated in the TWL simulation, as well as the normal depth simulation in which no data are
enforced at the downstream boundary. The MSL simulation tends to produce smaller variations and lower peaks as the
280 downstream boundary is forced by a constant water level. The overestimation in water level peaks by the TWL and normal
depth simulations is likely a result of the uncertainties in the channel topology in MOSART. Moreover, the diffusive wave
equation (eq. 1) simplifies the momentum transport by neglecting the inertia terms (local and convective). In the diffusive
wave method, because the flood wave is considered as subcritical and diffusive (Trigg et al., 2009), the water level is mainly
controlled by the upstream discharge. In low-lying rivers, while gravity and friction may not be the dominant forcing, the
285 inertial force related with velocity changes in space and time dominates the flow momentum. As such, the flood wave
propagation from the downstream boundary is underestimated in the backwater zone. As shown in Fig. 5, the improvement
of the TWL simulation in predicting water level is reduced towards upstream. This is not unexpected in a reduced-physics
river model (Hodges, 2013) because implied in the diffusive wave equation (eq. 1), the energy head at the downstream CBC
is created by the pressure gradient as a result of the variation in water surface but it is lost rapidly upstream due to the
290 increase in riverbed elevation.

The model evaluation results illustrate that a river model on a regionally-refined global mesh can represent backwater effects
at the basin scale when properly specified downstream CBC is used. Thus, the model can be used to further examine the
contribution of the backwater drivers. It should be noted that the 1D river models are by no means comparable to 3D
hydrodynamic models (e.g. models in (Gori et al., 2020) and (Zhang et al., 2020)) at reproducing coastal flood events or
295 resolving the complex flow dynamics. Therefore, our analysis focuses on a larger temporal scale by extracting the extreme
events from a long period, which are then used to quantify the backwater drivers.

## 4 Flood event simulation

The event selection method (Section 2.4) is used to select the extreme SS, LFS and FF events during Hurricane Irene,
Tropical Storm Lee and Hurricane Sandy from long-term observations at the NOAA coastal gauge (8534720), the NOAA
gauge closest to the river mouth (8545240) and the USGS gauge at Trenton (01463500), respectively (Fig. 7). A lag time of
4 hours is added to the water level data at the coastal gauge to compensate for the phase lag between this NOAA tidal gauge
and that at the river mouth. As there is an overlap between the SS and FF events during Irene (Fig. 7a), a compound flood is
identified over the combined period. The obvious difference is observed between the LFS and SS events, which were
obtained using the same method but at different locations. At the Delaware River mouth, the LFS event can be attributed to
both SS and river discharge, resulting in a duration period comparable to an FF event and much longer than a SS event. This





highlights the importance of considering the influence of high river discharge on compound flooding, particularly for rivers contributing to a small receiving water body. Sandy did not induce significant fluvial processes, and the LFS event was primarily caused by SS. Thus, no compound flood is identified over this period.

Because the TWL simulation shows a reasonable performance at reproducing both river discharge and water level during the

hurricane periods (Section 3), the TWL simulation is used to estimate the water depth change ($\Delta h$) and water volume change ($\Delta V$) in Eq (7) and (8) to assess the backwater effects. $\Delta h$ shows the backwater propagation extent, which is roughly 60 km upstream from the river mouth (Figure 7). This extent is determined by the riverbed elevation. Increasing the elevation to ~5 m, $\Delta h$ implies a large increase in the downstream water level. As a spatially aggregated quantity $\Delta h$, $\Delta V$ is also consistent with LFS, with $\Delta V$ following the LFS variation and peaking on the same dates. By comparing $\Delta h$ and $\Delta V$ over the two

hurricanes, it is not difficult to conclude that the compound flood caused by two consecutive events of Hurricane Irene and Tropical Storm Lee has a much larger impact on the backwater effects than that of Hurricane Sandy, even though their LFS peaks are at similar levels. This is probably because the duration of the SS event during Sandy is much shorter than the combined SS and FF events during Irene.

## 5 Backwater drivers

The interactive effects of water level and discharge inspire us to further decompose the backwater drivers. This section provides the analysis on the drivers based on the three-level decomposition introduced in Section 2.5. We examine the contribution of each driver to the backwater effects and their corresponding long-term trend under climate change.

### 5.1 Decomposition of backwater drivers

#### 5.1.1 Discharge, TWL and topology

The first decomposition level assesses the impacts of river discharge ($Q$), TWL and topology by comparing $\Delta V$ between Susquehanna River and Delaware River over the entire simulation period (Fig. 8). The result shows the key role of river topology and TWL in affecting backwaters. The maximum $\Delta V$ in Susquehanna River is roughly 3 orders of magnitude smaller than that in Delaware River. This is the result of a larger gradient in riverbed elevation profile of Susquehanna River that impedes the backwater propagation. Over the 5-km downstream section, the elevation increases from 0 to ~20 m in

Susquehanna River but by less than 1 m in Delaware River.

In both rivers, $\Delta V$ is dominated by TWL, with the corresponding correlation coefficient ($r$) over 0.9 (Fig. 9). However, the influence of $Q$ differs significantly between the two rivers. In Susquehanna River, $Q$ is negatively correlated with $\Delta V$ ($r = -0.11$). The increase in $Q$ reduces the TWL impact on $\Delta V$. For instance, at the same TWL, a smaller $Q$ could result in a higher $\Delta V$ (Fig. 8a). This behavior is also evident in the slopes of the fitted linear regression lines: the fitted slope for $Q \geq$

1000 m³/s is smaller than those for low discharge (Fig. 8a). This result is expected because high upstream discharge can



attenuate the propagation of downstream backwaters. In Delaware River, $\Delta V$ increases with $Q$ and $r$ between $\Delta V$ and $Q$ is 0.36. The regression slopes are similar at different discharge conditions. These contrasting results between the two rivers imply that the impact of $Q$ on the backwater effects depends on the river's receiving water body. Because Delaware River contributes to Delaware Bay, a much narrower estuary than Chesapeake Bay, the effect of its discharge on the water level

variation of the estuary is much stronger than that of Susquehanna River.  Consistently, there is a much higher $r$ between $Q$ and the TWL (0.27) in Delaware River than that (-0.03) in Susquehanna River (Fig. 9). In addition, the channel constriction in Delaware River might also facilitate the formation of backwaters (Castelltort et al., 2020).

Between the two draining estuaries, Chesapeake Bay behaves more like an ocean as the river impact is limited and the coastal and fluvial processes are distinct. Such situation is usually taken as the general case for compound effects and has

been addressed extensively using statistical models (Nasr et al., 2021; Ward et al., 2018) and large-scale river models (Ikeuchi et al., 2017). In contrast, cases like Delaware River have rarely been documented in any global-scale studies, even though they are ubiquitous and may witness more coastal backwaters. A possible reason for why such cases were overlooked is that previous global meshes have large deficiency in resolving narrow estuaries properly. Thus, we focus the following analysis on Delaware River.

**5.1.2 LFS and tide**

Given the key impact of TWL, the second decomposition level examines the respective role of the LFS and tide. The simulations configured with the downstream BCs of the TWL, LFS and tide (Section 2.6) are used to derive the maximum water depth change ($\Delta h_{max}$) for each grid cell:

$$\Delta h_{max}(i) = \max\{\Delta h(t, i): t = 1, 2, \ldots, T\}, \tag{9}$$

where $T$ is the simulation period.

The $\Delta h_{max}$ comparison shows the dominance of LFS over tide in increasing the maximum water depth (Fig. 10). In the along-channel profile (Fig. 10a), the value of $\Delta h_{max}$ in the LFS simulation is close to that in the TWL simulation: $\geq$1 m within the 40 km upstream range of the outlet and then gradually reduced to 0 with a sharp increase of the riverbed elevation. In contrast, the tide simulation produces a much smaller $\Delta h_{max}$ with the value never exceeding 0.5 m. Among the simulation

cases, the highest $\Delta h_{max}$ occurs at roughly 25 km upstream from the mouth. This along-channel profile reveals the interaction of the discharge and the upstream propagation of tide and surge momentum. It is also observed that $\Delta h_{max}$ is slightly higher in the LFS simulation than in the TWL simulation, which is likely the result of the negative impact of low tide on TWL. The spatial variation of $\Delta h_{max}$ is shown in Figure 10b. The backwater effects are limited to the low-lying section of Delaware River, i.e. the downstream of Trenton. Backwater propagation occurs along the main channel as well as

some small contributing tributaries, for which the extent is determined by the corresponding elevation. The propagation extent is similar between the TWL and LFS simulations and is much smaller in the tide simulation.





### 5.1.3 High discharge, SS and compound effect

The LFS impact is further decomposed into high discharge, SS and their compound effect using the LFS simulation. We compared the variation of the event accumulated $\Delta V$ with respect to the event duration and peak water level among the

drivers (Fig. 11). Regardless of the drivers, $\Delta V$ is mainly determined by the event duration. Its value linearly increases with the duration with high correlations. The peak water level provides the secondary effect. Higher peaks generally increase $\Delta V$, resulting in values above the fitted regression line.

Our result indicates that the influence of each driver on $\Delta V$ is more dependent on the frequency and duration of the corresponding events rather than their extremes. For example, the FF events are more influential on $\Delta V$ because they are

more frequent than the SS and compound flood events. Also, the $\Delta V$ is higher in the FF and compound flood events than in the SS events because the latter lasts much shorter. A remarkable case occurred during Hurricane Irene when the highest $\Delta V$ in our study period was produced by the combined long-lived FF and compound flood events (Fig. 7). Noticeably, the slope and correlation between $\Delta V$ and duration is the largest in the compound events, which means that the strength of the compound events increases more rapidly with duration. In all, our driver comparison indicates that high discharge is the key

driver of the backwaters in Delaware River due to the higher frequency and longer duration of the corresponding FF events.

### 5.2 Trend analysis

This section performs an analysis of annual trend on the LFS and decomposed drivers, as well as their influences on the backwater effects. The result reveals the impacts of SLR and increasing frequency of SS events during 1990-2019 in exacerbating the backwater effects.

### 5.2.1 Trend in the backwater drivers

The annual trend of the backwater drivers shows an increasing trend of SS due to both SLR and increasing SS frequency (Fig. 12), with $p$-value as 0.039 and 0.031, respectively, for annual duration and occurrence in the SS+SLR case (Table 1). When SLR is considered, the number of SS occurrence increases from ~1 to 2~5 times per year over the study period (Fig. 12b). Accordingly, the annual duration of the SS event increases from ≤15 days (1990-2005) to >25 days over multiple

years (2005-2019) and is up to over 50 days in 2016 (Fig. 12a). The SS peaks are also increased by SLR but do not present any trend (Fig. 12c). Neither do we notice clear trends from the FF events nor the compound events (Table 1). The annual characteristics of the FF events vary significantly between wet and dry years. In the very wet year of 1996, the duration of the FF events reaches 100 days with up to 6 events per year. The frequency of compound flood events is low, only occurring 1~3 times per year between 2003 and 2012.

Affected by the drivers of high discharge and SS, the LFS events show an increasing trend in both duration and frequency with $p$-value at 0.094 and 0.042, respectively. No clear trend is found in the LFS peaks (Table 1). The LFS trend is basically consistent with that of the SS events with a few exceptions in flood years (i.e., year 1996 and 1998) when the LFS events



were caused by high discharge. Except for the flood years, the LFS occurrence is as low as one time per year in the 1990s and increases to 2~6 times per year since 2003. Correspondingly, the LFS duration increases from ≤5 to 20~40 days since 2004. The increased trend in LFS is likely a result of SLR that leads to more frequent occurrences of the SS events.

### 5.2.2 Trend in the backwater effects

The annual trend of the backwater effects is analyzed for the different drivers (Table 1) in terms of the annually accumulated $\Delta V$ (Fig. 13). The trend of $\Delta V$ is consistent with the event duration and frequency trends of the corresponding drivers. The SS and LFS induced backwater effects are increasing but no clear trends can be observed for the FF and compound flood induced backwater effects. The $\Delta V$ trends are significant in the SS+SLR, SS and LFS cases and insignificant in the FF and compound cases (Table 1). Our result also demonstrates the critical impact from high discharge. The resulting $\Delta V$ by FF over the flood years (e.g., year 1996) can be several times higher than the $\Delta V$ caused by the other drivers. This is likely due to the long duration of the FF events. Because SLR and intensified SS increased coastal backwaters in river channels, our analyses call for better representations of the related processes in ESMs for predictive understanding of associated flood risks under climate change and effects on the water and biogeochemical cycles through land-river-ocean interactions and possible impacts on atmospheric processes. However, we caution attributing changes based on modeling and analysis of a 30-year period to climate change as internal climate variability and other anthropogenic effects likely also play important roles in the increasing sea level and storm surge frequency.

### 6 Discussion

Our study shows that using the diffusive wave method, large-scale river models configured on a coastal refined mesh are capable of reproducing backwater effects in low-lying river channels with appropriate downstream boundaries. The global unstructured mesh alleviates the computational burden in ESMs by relaxing the resolution in the inland and offshore regions to ~100km while embedding regions of "high" resolution of O(1km) near the river-ocean interface. Although the resolution is still not comparable to that used in local-scale models, the mesh is able to resolve the complex river networks near the coastline without having to merge multiple outlets into a single cell. The downstream boundary condition is critical for connecting the coastal and fluvial processes, transferring the water head energy upstream and thus simulating the backwater effects. The widely used diffusive wave method that uses the normal depth boundary and the more simplified kinematic wave method may be only applicable in high-gradient regions, as these methods do not incorporate any downstream information.

As an important finding, this study revealed the crucial difference in flood drivers between two distinct coastal rivers (i.e., Susquehanna River vs. Delaware River), with the former connected to a wide ocean-like estuary and the latter connected to a narrow river-like estuary, which is usually ignored by ESMs. The difference is mainly caused by the effects of river geometry and estuary size on LFS which is a direct driver of backwaters. For an ocean-like estuary, such as Chesapeake Bay,





river discharge from its drainage basins hardly affects the water level fluctuations of the estuary. But when a coastal river

drains to a narrow estuary, its LFS would be driven by not only storm surge but also river discharge and their compounding. Further, the backwater effects created by high discharge and storm surge are different. While storm surge generates an upstream-propagated energy head, high discharge gradually builds up the water level of the receiving water body. The increased water level would slowly move upstream, attenuating the river stage fluctuation and flood waves. High discharge that presents a higher frequency and a longer duration can occur in close successions with storm surge during compound

flooding, e.g., Hurricane Irene and Tropical Storm Lee, creating extended backwater effects. In all, we show that in addition to the conventional flood drivers, such as riverbed elevation and sea level, ESMs need to properly represent the flood drivers for small estuaries, such as river discharge.

We demonstrated that the backwater effects are significant in the low-lying watersheds and have an increased trend over the recent 30 years. In Delaware River, the propagated backwaters account for up to 1.2 m increase in water depth and $1\times10^8$ m$^3$

increase in water volume per day during an extreme event. The effects could be several orders higher for larger river basins. This increased flood risk will otherwise be underestimated if the backwater effects are not properly represented in ESMs. Furthermore, our simulation also shows the increased influence of climate change on backwaters, with SLR and more frequent storm surge increasing the strength of backwaters in the mid-Atlantic region.

Noticeably, there are still a few limitations in the river model used in this study, which may introduce uncertainties to our

simulations. First, as a large-scale river transport model, MOSART simplifies the channel cross-section as rectangular and trapezoidal when the water depth is below and above the river's bankfull depth, respectively, partially due to lack of large-scale river geometry data (Li et al., 2013). This simplification may affect the accuracy of simulated water depth for rivers with very irregular channel cross-section. Second, while we demonstrated that the direct impact of tide on the backwater effects is limited compared to LFS, the quantification assumes that LFS and tide can be separated by removing tide from the

total water level, ignoring the nonlinear interaction between tide and surge. In reality, as a dynamic component in Delaware Bay, the interaction between high tide and coastal surge may stimulate a further increase in the water level (Krien et al., 2017). Last but not least, our simulation takes the in-situ observation as the boundary data, which does not account for the interaction between fluvial and coastal processes. However, the river-ocean interface is a dynamic region that features complex multiscale processes. In the context of an intense storm, the upstream propagation of storm surge impedes river

discharge, which in turn modulates the water level. It remains unclear if this mutual interaction is critical in flood modeling. Thus, we aim to couple MOSART with the E3SM ocean model interactively in our next step for improving predictive understanding of coastal floods.

## 7 Conclusion

This research assesses the capability of the global-scale MOSART river transport model to simulate the coastal backwater

effects at the basin scale by imposing the observed water level at the downstream boundary and using a coastal refined



unstructured mesh. The simulation is evaluated at two major river basins of the densely populated mid-Atlantic region. MOSART shows a reasonable agreement with the observed river discharge in both rivers and captures the water level variations in the downstream section of Delaware River, indicating the model's capability in representing the backwater effects. We performed numerical experiments and extracted extreme flood events to examine the contribution of various

backwater drivers. Our analyses revealed the dependence of the backwater drivers on the river geometry and the river's receiving water body. While storm surge is considered a dominant forcing, the impact of high discharge can be significant in a narrow river-like estuary, such as Delaware River, as the discharge modulates the low-frequency water level variations within the estuary. In addition, high discharge when occurring simultaneously with storm surge, creates strong compound flooding. The extreme impact of a compound event should be mainly attributed to the extended duration of the combined

coastal and fluvial processes rather than extreme flood peaks. The trend analysis of the backwater drivers shows that the strength of backwaters in Delaware River has been increasing in the recent decades due to SLR and more frequent storm surge. In the future, we plan to extend the current work from the mid-Atlantic region to the global domain and refine the coastal mesh globally. A framework of two-way coupled river and ocean models will be established to understand the complex river-ocean interactions.

**Code and data availability**

The water level observation can be assessed though Center for Operational Oceanographic Products and Services (CO-OPS, https://tidesandcurrents.noaa.gov/). The river discharge data are obtained from USGS National Water Information System (https://waterdata.usgs.gov/nwis).  The  developed  MOSART  code  is  available  at https://github.com/fdongyu/E3SM/tree/fdongyu/mosart_tide.

**Author contributions**

DF and ZT designed the experiments. DE, CL and DX prepared the model mesh and parameters. DF carried out the analysis. DF and ZT wrote the initial draft of the manuscript. All authors contributed to the editing of the paper.

**Competing interests**

The authors declare that they have no conflict of interest.

**Acknowledgements**

The work presented in this manuscript is supported by the Earth System Model Development program areas of the U.S. Department of Energy, Office of Science, Office of Biological and Environmental Research as part of the multi-program,



collaborative Integrated Coastal Modeling (ICoM) project. PNNL is operated for DOE by Battelle Memorial Institute, United States under contract DE-AC05-76RL01830. All model simulations were performed using resources available
through Research Computing at Pacific Northwest National Laboratory.

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

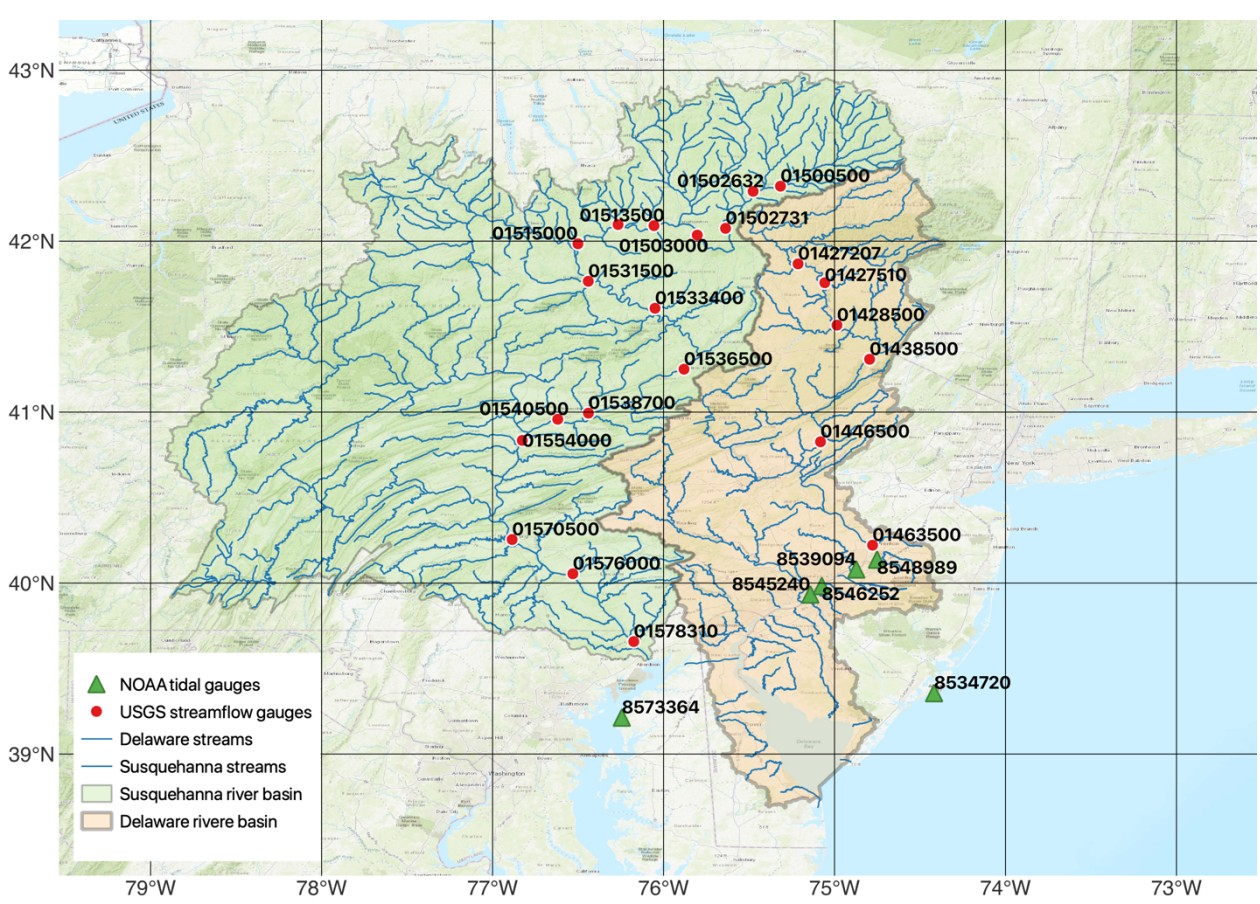

**Figure 1: An overview of Susquehanna River Basin (SRB) and Delaware River Basin (DRB). This map is created using the free and open source QGIS on the world topographic map (ESRI, 2012).**

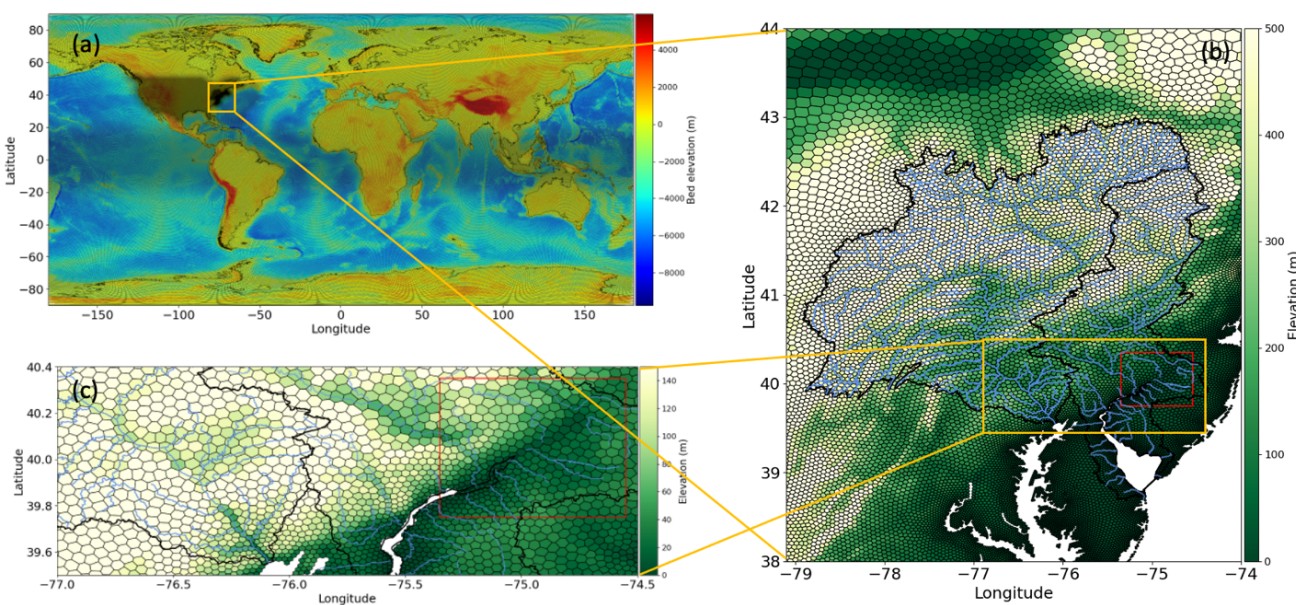

**Figure 2: (a) The global unstructured mesh of E3SM. (b) A magnified view of the mid-Atlantic region (orange rectangle in (a). (c) A magnified view near the mouths of Susquehanna River and Delaware River (orange rectangle in (b). The red rectangles in (b) and (c) represent the downstream section of Delaware River used for backwater propagation analysis.**

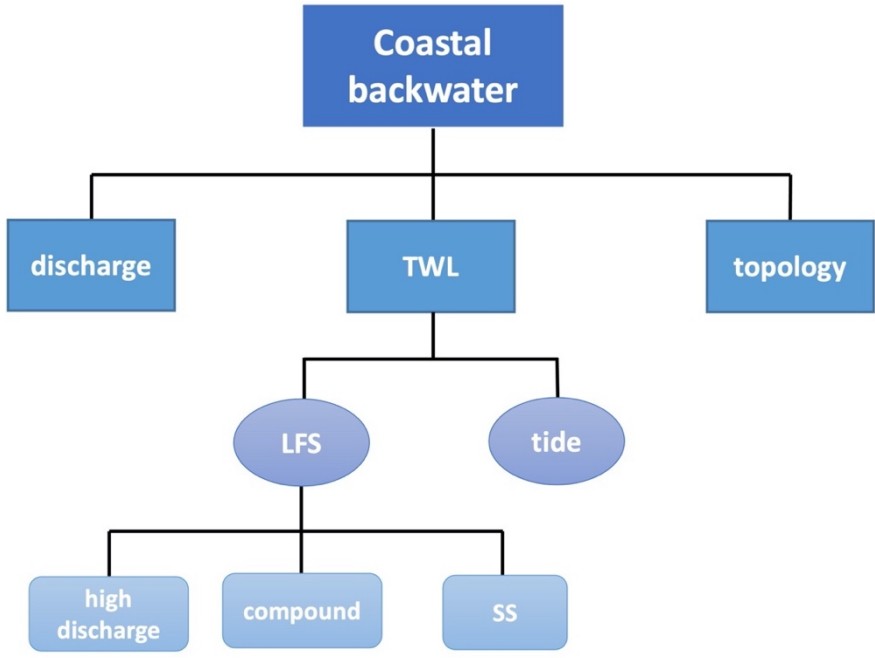



**Figure 3: Decomposition of the drivers of backwater effects. TWL is total water level, LFS is low-frequency surge, and SS is storm surge.**

**Figure 4: The river discharge evaluation in SRB and DRB: (a) $r^2$, (b) KGE, (c) hydrograph at USGS gauge 01463500, (d) hydrograph at USGS gauge 01578310. The triangles and the circles in (a) and (b) represent the USGS gauges in the main stem of Susquehanna River and Delaware River, respectively.**



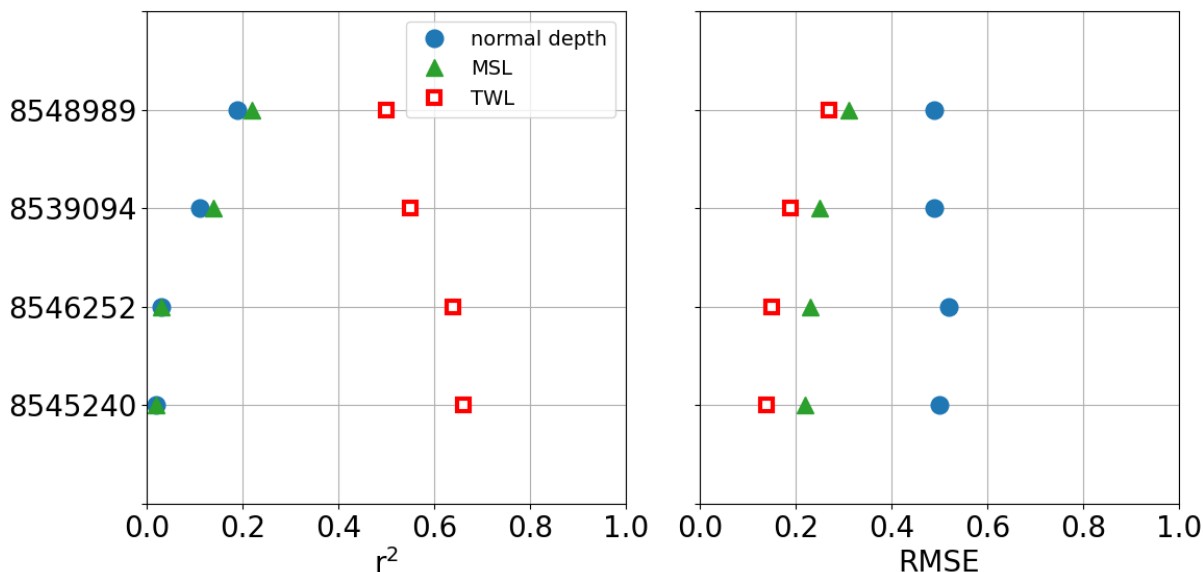

**Figure 5: The $r^2$ and RMSE of the MOSART simulations at the 4 NOAA gauges.**

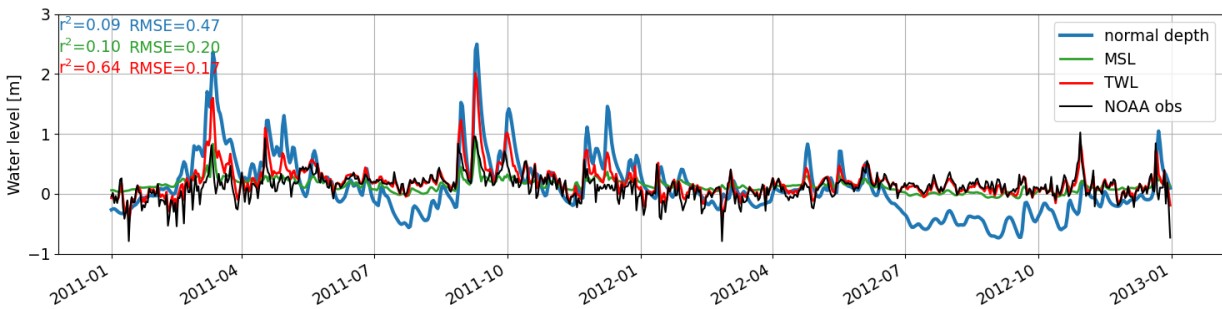


**Figure 6: Comparison of simulated and observed water level at NOAA gauge 8545240.**



**Figure 7: The top panels are the three types of selected extreme events (SS, LFS, FF) overlaid on the corresponding time-series observations (grey solid and dashed lines) for Hurricane Irene and Tropical Storm Lee (a) and Hurricane Sandy (b). The compound flood event is marked between the two black vertical lines. The bottom left panels are the riverbed elevation along the backwater propagation extent in Delaware River and the bottom right panels are Δh (color shading) along the upstream distance and ΔV (black curve) derived from the MOSART simulations.**




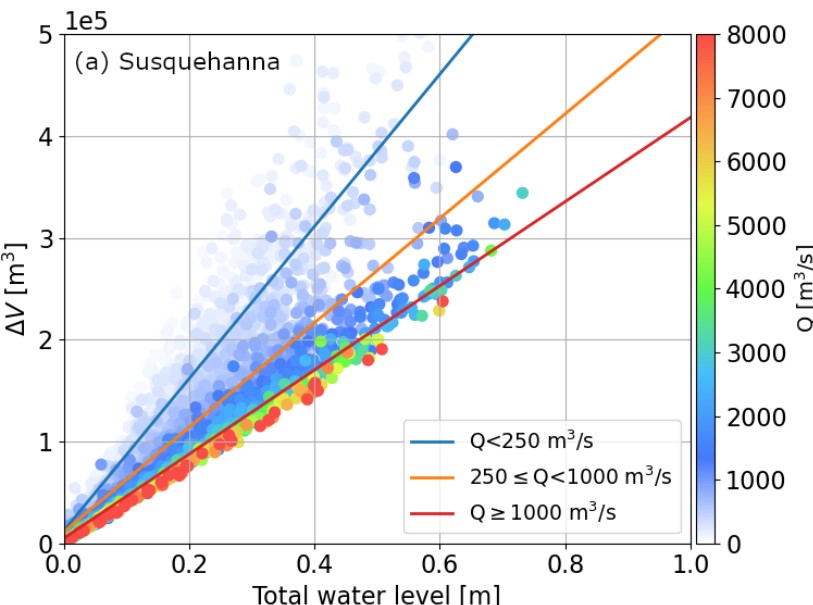

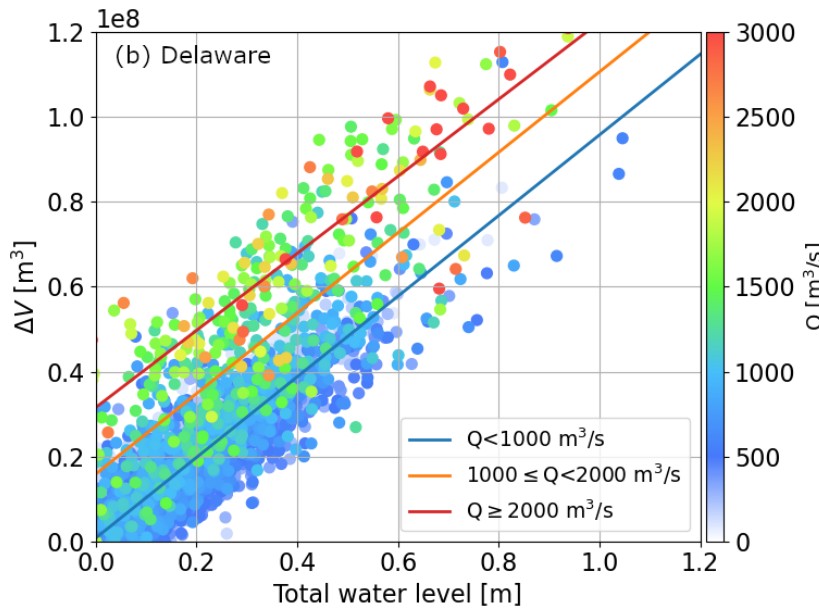

**Figure 8: Scatter plot of ΔV against TWL in Susquehanna River (a) and Delaware River (b). Colored circles represent the corresponding river discharge (Q). The TWL and Q data are respectively obtained from gauges 8573364 and 01578310 for Susquehanna River, and from gauges 8546252 and 01463500 for Delaware River. The solid lines are the fitted linear regression under different discharge ranges.**



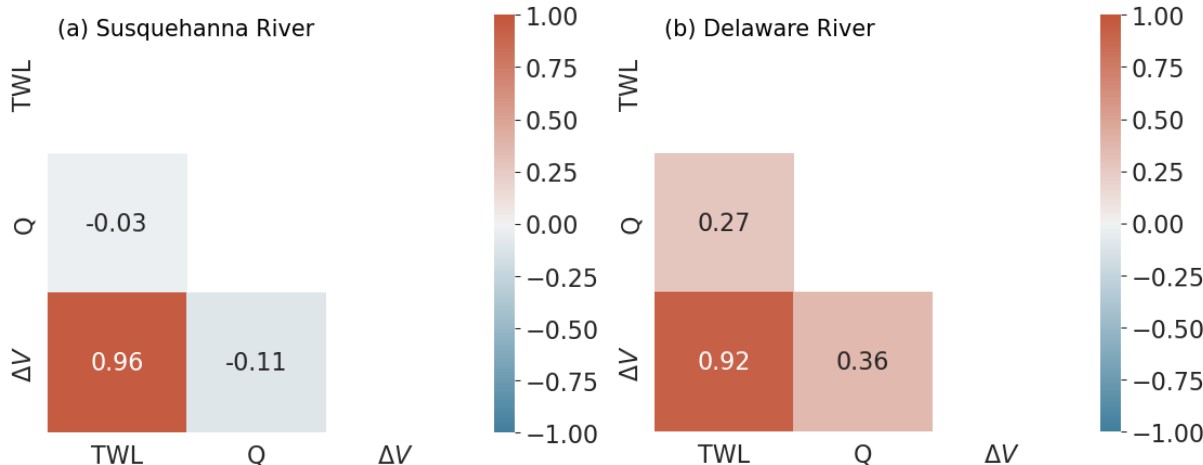

**Figure 9: The correlation coefficient (r) matrix of $\Delta V$ , $Q$ and TWL in Susquehanna River (a) and Delaware River (b).**


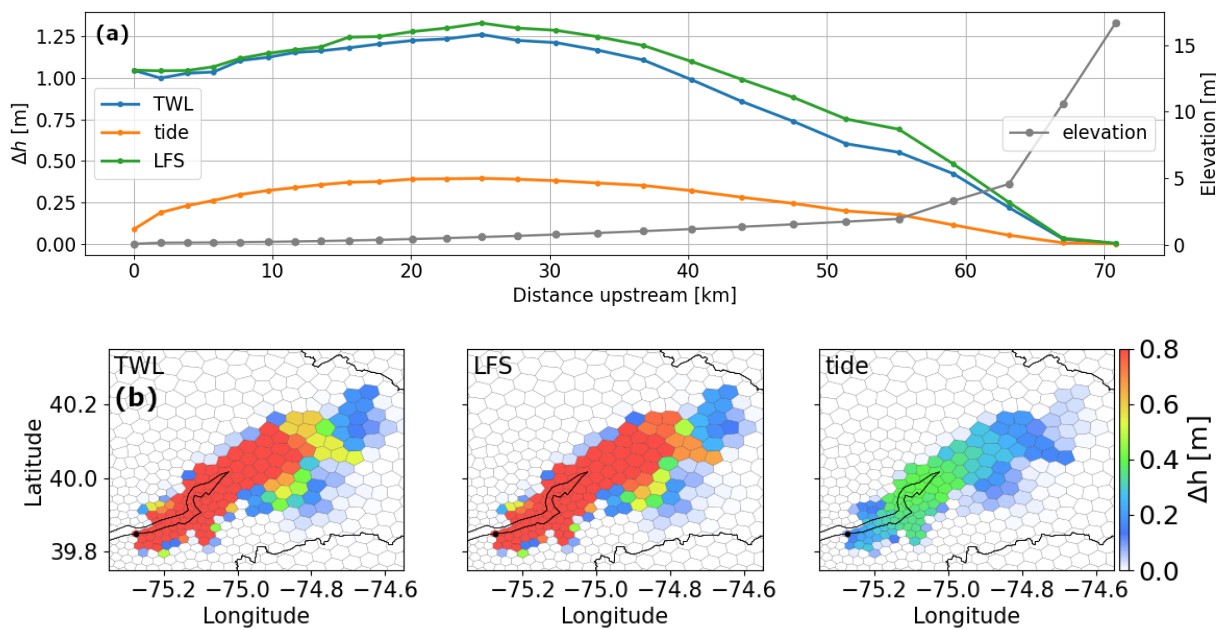

**Figure 10: (a) The along channel profiles of the maximum $\Delta h$ obtained using the TWL, SS and tide configurations. The river outlet is at x = 0 km. (b) The corresponding spatial map of the maximum $\Delta h$ over a downstream region of Trenton. This region is specified as the red rectangle in Figure 2. The black dots represent the Delaware River outlet.**


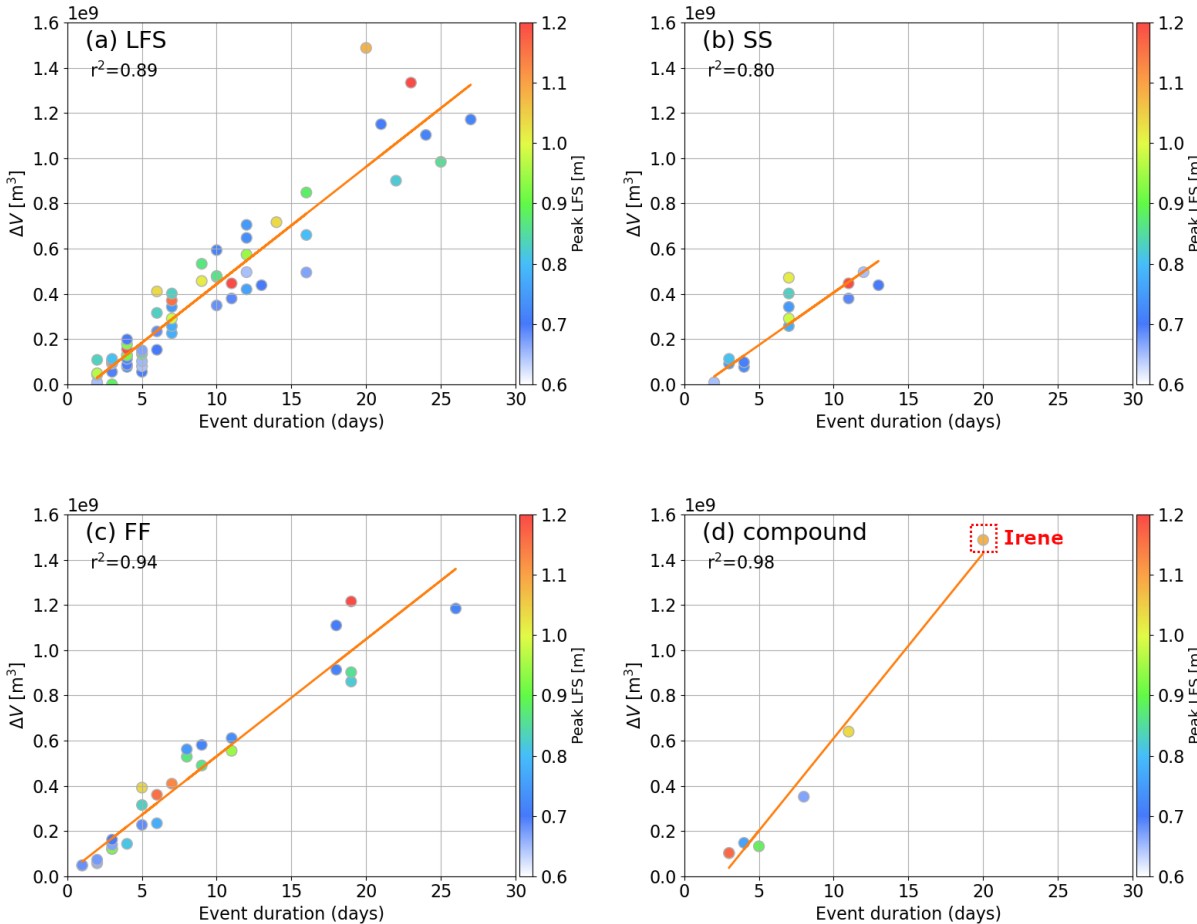

**Figure 11: Scatter plot of the accumulated ΔV against the event duration in days for the LFS events (a), the SS events (b), the FF events (c) and the compound flood events (d). The color represents the corresponding peak LFS.**



**Figure 12: The interannual variability of (a) the event annual duration, (b) the event occurrence and (c) the maximum peak value for different types of flood events.**





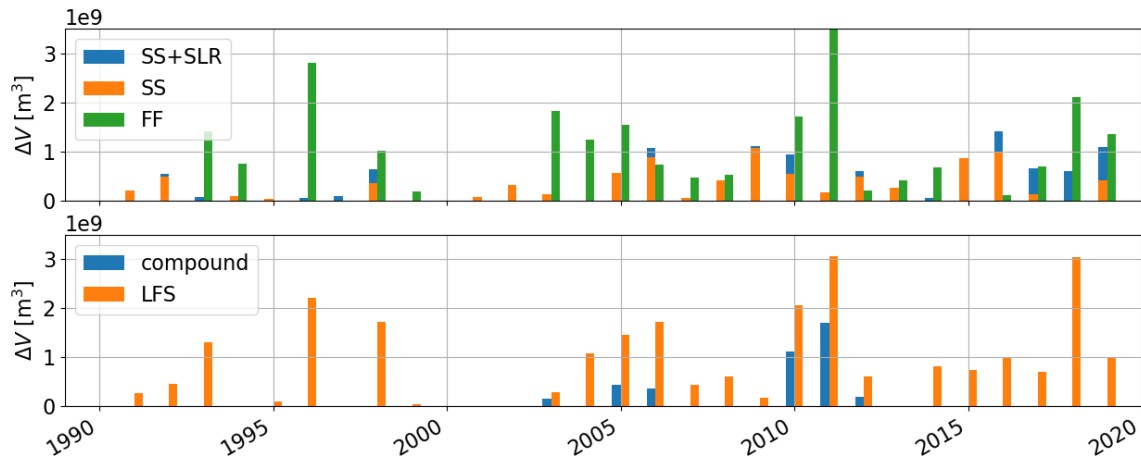

**Figure 13: Similar to Figure 12 but for the interannual variability of $\Delta V$.**

**Table 1: The MK statistics of long term trends in annual duration, occurrence, peak value, and $\Delta V$.**

|  |  | SS+SLR | SS | FF | compound | LFS |
|---|---|---|---|---|---|---|
| Annual duration | $Z$ | 2.064 | 0.809 | 0.794 | -0.091 | 1.674 |
|  | $p$-value | 0.039 | 0.419 | 0.427 | 0.928 | 0.094 |
| Occurrence | $Z$ | 2.157 | -0.286 | 0.728 | 0.976 | 2.032 |
|  | $p$-value | 0.031 | 0.775 | 0.467 | 0.329 | 0.042 |
| Peak value | $Z$ | 0.446 | -0.986 | 0.361 | 0.968 | 0.950 |
|  | $p$-value | 0.656 | 0.324 | 0.718 | 0.333 | 0.342 |
| $\Delta V$ | $Z$ | 2.720 | 1.900 | 1.286 | 0.742 | 1.900 |
|  | $p$-value | 0.007 | 0.057 | 0.199 | 0.458 | 0.057 |