# Peer review of "Investigating coastal backwater effects and flooding in the coastal zone using a global river transport model on an unstructured mesh"

_Hydrology and Earth System Sciences, 2022_

## Referee Comment (RC1)

Review of "Investigating coastal backwater effects and flooding in the coastal zone using a global river transport model on an unstructured mesh" by Feng et al.

In this study, the authors explored the coastal backwater effects and flooding in the coastal zone making use of a global river transport model. Here the backwater effect was defined as an extreme phenomenon when the downstream river stage is higher than the river stage of the current channel, resulting in the negative friction slope and hence negative flow velocity. In particular, the backwater effects were mainly induced by the dynamic sea level variation, especially during the storm surge event together with the fluvial flood. Generally, the paper is well organized and written. However, there are still some major and minor concerns that should be properly addressed.

**Major concerns:**

1. The valid of decomposition method applied to the river mouth (i.e., 8545240 in Figure 1): It can be seen from Figure 1 that the tidal gauge with no. 8545240 is located in the upstream part of the Delaware Bay (or Delaware river delta), thus the water level fluctuation is not featured by typical tidal cycles (either semidiurnal cycle or spring-neap cycle, see also Figure 6 in the manuscript). Consequently, my major concern is that the decomposition of the total water level (TWL) into low-frequency surge (LFS) and tide is still valid in this station (i.e., 8545240), especially during the flood events when the river discharge is substantially larger than the tidal discharge since in this case the tidal signal is weak and hard to be detected.

2. The missing of realistic tide-river interaction and its associated backwater effects: As presented in Section 2.2, the global river routing model is mainly driven by runoff from a land surface model. This implies that the model did not explicitly account for the potential impacts of tide from the open sea. Thus, in principle, the adopted MOSART model intuitively neglects the tide-river interaction, which is one of the major physical mechanism leading to backwater effects in coastal zones. For the time being, the backwater effects in this study is mainly constrained by the upstream river discharge and downstream total water level, without explicitly accounting for the tide-river interaction.

3. Figures 4 and 6: In Figure 4, we observe that MOSART model significantly underestimated the extreme flood peak. Thus, my major concern is whether the two computed quantification metrics (including water depth change  $\Delta h$  and water volume change  $\Delta V$ ) for backwater effects are reliable, especially during the Hurricane Irene and Hurricane Sandy. The case is similar for the reproduction of the water level in tidal gauge of 8545240. As a result, the reliability of computed  $\Delta h$  and  $\Delta V$  should be clarified. By the way, the current MOSART model is also driven by the total water level observed in tidal gauge with no. 8545240, i.e., as a coastal boundary condition? If so, then the comparison between observed and computed water levels presented in Figure 6 is not proper.

4. The negatively correlated relationship between river discharge Q and  $\Delta V$  (or Q and total water level TWL) in the Susquehanna River has to do with the fact that the imposed total water level was derived from the tidal gauge of 8545240, which is located

in the Delaware River? Please clarify the adopted coastal boundary condition adopted in the Susquehanna River basin.

5. Since most cities are actually located in the downstream part of the adopted coastal boundary condition (e.g., 8545240 in Figure 1), the real application of flood control from this study focusing on the river basin is questionable. Further explanation is required for the implication of this contribution.

Some minor comments:

1. Section 2.1: the study domain does not belong to Methodology.

2. Lines 129-130: Please also illustrate the backwater effect in two studied regions.

3. Line 257: the definition of Kling-Gupta efficiency (KGE) is missing.

4. Lines 282 and 288: eq. 1→Eq. 1

5. Line 403: How did you compute the time series of  $\Delta V$  (for the whole river basin or only in the main channel)?

6. Figure 7: What are the observations for the grey solid and dashed line? In addition, please correct the units for the  $\Delta V$  and the discharge

7. Line 476: though  $\rightarrow$  through

---

## Author Comment (AC1)

Response to Reviewers

Title: Investigating coastal backwater effects and flooding in the coastal zone using a global river transport model on an unstructured mesh

Author Response 1st revision

Reviewer 1
Reviewer Comments:

In this study, the authors explored the coastal backwater effects and flooding in the coastal zone making use of a global river transport model. Here the backwater effect was defined as an extreme phenomenon when the downstream river stage is higher than the river stage of the current channel, resulting in the negative friction slope and hence negative flow velocity. In particular, the backwater effects were mainly induced by the dynamic sea level variation, especially during the storm surge event together with the fluvial flood. Generally, the paper is well organized and written. However, there are still some major and minor concerns that should be properly addressed.

Author Response:

We appreciate the reviewer for the critical assessment of our work. The comments and suggestions are insightful and helpful. In the following we address your concerns point by point. All changes in the revised paper have been marked with blue color. If you have any further questions or concerns, please let us know. Although HESS does not allow us to share our revised manuscript at this stage of the review process, we provide excerpts throughout the response to help illustrate the changes. We will provide the revised manuscript when invited. Please note the line numbers are according to the revised version.

Major concerns:

R1C1:

1. The valid of decomposition method applied to the river mouth (i.e., 8545240 in Figure 1): It can be seen from Figure 1 that the tidal gauge with no. 8545240 is located in the upstream part of the Delaware Bay (or Delaware river delta), thus the water level fluctuation is not featured by typical tidal cycles (either semidiurnal cycle or spring-neap cycle, see also Figure 6 in the manuscript). Consequently, my major concern is that the decomposition of the total water level (TWL) into low-frequency surge (LFS) and tide is still valid in this station (i.e., 8545240), especially during the flood events when the river discharge is substantially larger than the tidal discharge since in this case the tidal signal is weak and hard to be detected.

Author Response:

We appreciate the thoughtful comment and apologize for not being clear at this point. We agree that the dominance of tides is a prerequisite for harmonic tidal analysis at this level of decomposition.

In fact, Station 8545240 (https://tidesandcurrents.noaa.gov/waterlevels.html?id=8545240) is a tidal gauge with strong tidal signals and typical tidal cycles. This is confirmed in the study of Xiao et al., 2021 (10.3389/fmars.2021.715557). In their study, the total water level at this station was decomposed to harmonic tide and low frequency surge using a similar way. Please see their Figure 8 that shows an example of a few tidal cycles over Hurricane Irene and Tropical Storm Lee (their station Philadelphia is station 8545240). Even at high-flow conditions during Irene and Lee, the tidal effects are still significant. In this study, Figure 6 compares the daily-averaged water level, as the daily scale is the typical time scale for hydrological models.

In response to this comment, we have emphasized that the measurable tidal effect is required when decomposing total water level to tide and low-frequency surge (in section 2.5, Line 211-213):

"It should be noted that this level of decomposition must be applied to a tidal gauge as the harmonic tidal analysis requires measurable tidal effects."

We provided more information of Station 8545240 (Section 4, Line 318-320):

"Gauge 8545240, despite located at the upstream reach of Delaware Bay, is dominated by semidiurnal tides even at high-flow conditions during extreme storm events (Xiao et al., 2021)."

We also clarified "daily-averaged" in the caption of Figure 6. According to the comment R1C3, we now show the water level comparison at Station 8546252. Please see the response below.

R1C2:

2. The missing of realistic tide-river interaction and its associated backwater effects: As presented in Section 2.2, the global river routing model is mainly driven by runoff from a land surface model. This implies that the model did not explicitly account for the potential impacts of tide from the open sea. Thus, in principle, the adopted MOSART model intuitively neglects the tide-river interaction, which is one of the major physical mechanism leading to backwater effects in coastal zones. For the time being, the backwater effects in this study is mainly constrained by the upstream river discharge and downstream total water level, without explicitly accounting for the tide-river interaction.

Author Response:

We appreciate the reviewer's insights and agree that mutual tide-river interaction is one of the major physical mechanisms on backwater effects. However, even though the interactively-coupled river-ocean models may be used to account for the dynamic interaction, the influence

of tide, surge and their interaction is still considered in the large-scale river model with the prescribed time-varying downstream boundary (Ikeuchi et al., 2017; Eilander et al., 2020), especially as we refined the mesh properly to propagate the coastal processes upstream. This is also implied in Figure 5 that the model achieves satisfied accuracy in the few tidal gauges.

Even without the interactively coupling, the model development in this study has certain values. As already stated, the new model development makes it possible to investigate coastal flooding in the context of climate change as the global river-ocean coupled simulation will be computationally expensive. Compared with the previous one-way river-ocean coupling in ESMs, the new coupling does improve the representation of backwater effects in low-lying coastal regions. And these regions are likely also the regions that are more prone to coastal and compound flooding. So even the tidal propagation is underestimated, the model will still be very useful to provide insights on the relative change of coastal flooding in the context of climate change. Below, we provide the existing discussions, based on which we elaborate more according to the reviewer's suggestions.

In the original manuscript, we have specified the advantages of large-scale models and acknowledged the existing limitations:

As discussed in the introduction, we show that there are growing applications of large-scale river models (similar to MOSART) to assess compound flooding at various scales by coupling with the downstream total water level (Chen et al., 2013; Ikeuchi et al., 2017; Yamazaki et al., 2012; Eilander et al., 2020; Mao et al., 2019), because the large-scale river models are more computationally efficient and can be coupled directly with other components of earth system models (Line 69-72) to address the change of regional and global water and biogeochemical cycles (Line 88).

In the model evaluation (Section 3.2, Line 297-302), we acknowledge that the tidal propagation in the backwater zone could be underestimated due to the simplified physics of MOSART, which is a typical challenge in large-scale river modeling, and that "the 1D river models are by no means comparable to 3D hydrodynamic models at reproducing coastal flood events or resolving the complex flow dynamics (Neal et al., 2012)". Large-scale river models are more important for a large spatiotemporal scale assessment in a changing climate.

In the discussion (Line 482), we discuss the current limitation for not considering river-tide interaction in the simulation: "because our simulation takes the in-situ observation as the boundary data, it does not account for the interaction between fluvial and coastal processes. In reality, the river-ocean interface is a dynamic region that features complex multiscale processes. In the context of an intense storm, the upstream propagation of storm surge impedes river discharge, which in turn modulates the water level". However, "it remains unclear whether this mutual interaction is critical in flood modeling".

We also demonstrate the ongoing development of coupling "MOSART with the E3SM ocean model interactively to improve predictive understanding of coastal floods", as part of a larger

effort to develop capabilities in representing land-river-ocean interactions in E3SM (Line 86). We believe that the only possible way to include tide-river interaction is to have an active ocean component in the coupling framework given the fact that there is no direct measurement of upstream ocean fluxes.

In response to this comment, we elaborated more on the motivation of using a large-scale river model in the introduction (Line 72) and the limitation for neglecting the tide-river interaction and the ongoing work in the discussion (Line 482):

"Although hydraulic or hydrodynamic models were used more often in previous studies to simulate storm surge induced coastal inundation (Bakhtyar et al., 2020; Muñoz et al., 2020), there have been growing applications of large-scale river models to assess the compound fluvial and coastal flooding at basin (Chen et al., 2013), regional (Ikeuchi et al., 2017; Yamazaki et al., 2012) and global scales (Eilander et al., 2020; Mao et al., 2019) because they are more computationally efficient for a large spatiotemporal assessment. The long-term evolution of flood drivers and risks can be quantified in the context of climate change. Moreover, such models, when directly coupled with other components of ESMs, can also provide estimations of energy, biogeochemical and sediment processes that are often neglected in pure flood inundation models (Li et al., 2022). "

"Last but not least, because our simulation takes the in-situ observation as the boundary data, it does not account for the interaction between fluvial and coastal processes. The backwater effects are constrained by the prescribed boundary sea level. However, the river-ocean interface is a dynamic region that features complex multiscale processes. In the context of an intense storm, the upstream propagation of storm surge impedes river discharge, which in turn modulates the water level (Dykstra & Dzwonkowski, 2020). It remains unclear if this mutual interaction is critical in flood modeling and how it responds to sea level rise (Kulp & Strauss, 2019) and enhanced tidal dynamics (Talke & Jay, 2020) due to climate change. Thus, this study provides a basis for modeling coastal induced flooding in river basins. We aim to couple MOSART with the E3SM ocean model interactively in our next step for resolving the complex interactions at the river-ocean interface and eventually couple all land-river-ocean processes within E3SM to improve predictive understanding of compound flooding through pluvial, fluvial and coastal processes (Xu et al., 2022)."

R1C3:

3. Figures 4 and 6: In Figure 4, we observe that MOSART model significantly underestimated the extreme flood peak. Thus, my major concern is whether the two computed quantification metrics (including water depth change Δh and water volume change ΔV) for backwater effects are reliable, especially during the Hurricane Irene and Hurricane Sandy. The case is similar for the reproduction of the water level in tidal gauge of 8545240. As a result, the reliability of computed Δh and ΔV should be clarified. By the way, the current MOSART model is also driven by the total

water level observed in tidal gauge with no. 8545240, i.e., as a coastal boundary condition? If so, then the comparison between observed and computed water levels presented in Figure 6 is not proper.

Author Response:

Thanks for the comment.

As the reviewer noted and we discussed in Section 3.1, the simulated streamflow significantly underestimates the flood peak during Hurricane Irene. We attribute this error to uncertainty in the global runoff forcing (Section 3.1, Line 280). In this study, global runoff forcing was used to perform global river simulations. Even though the analysis focuses on the specific river basins using regionally refined mesh, the projected goal is to generalize the analysis to the global configuration. The global runoff forcing, although is bias corrected, may underestimate the extremes over specific regions. This is a known challenge in global runoff generation schemes. Another reason as we mentioned in Section 3.1 (Line 284) is the uncertainty from the USGS measurement. The streamflow that is estimated from the discharge-stage empirical relationship (rating curve) may be biased particularly during extreme events because these measurements occur less frequently (Turnipseed & Sauer, 2010). Despite the underestimation during Irene, our model captures the other peaks in Figure 4 and achieves overall satisfied performance based on a general criterion (Towner et al., 2019). Additionally, we specify that the analyses of the two example events are only used to "demonstrate the selection of extreme events and the quantification of the backwater effects", as stated in section 2.7 (Line 252).

To increase the clarity of the revised manuscript, we first elaborated the uncertainty source of the modeled discharge in Section 3.1:

"This bias is likely caused by uncertainty in the GRFR runoff forcing (Yang et al., 2021). The global runoff forcing, although is bias corrected, may underestimate the extremes over specific regions. This is a known challenge in global runoff generation schemes. Additionally, there also exists uncertainty in USGS discharge data estimated from rating curve during extreme events (Di Baldassarre et al., 2012), because these measurements occur less frequently and usually do not cover extreme events (Turnipseed & Sauer, 2010)."

We then emphasized the purpose of the event simulations at the beginning of Section 2:

"Two hurricane events are selected to demonstrate the applicability of the proposed methods."

Finally in the discussion, we clarified the reliability of computed Δh and ΔV (Line 475-478):

"Second, the global runoff forcing may underestimate the event extremes, such as the discharge peak during Hurricane Irene, affecting the reliability of backwater quantifications during the corresponding event. As we target at generalizing the analyses to the global scale, the bias-corrected global forcing that can capture extremes is desired."

The comparison in Station 8545240 was shown because this station is nearest to the outlet and has the data available over year 2011 and 2012 when Hurricane Irene and Sandy occurred. We understand the reviewer's concern at this point and now show the comparison in Station 8546252 from 2016 to 2017. This change does not affect the result discussions.

[Figure]

Figure 6: Comparison of daily-averaged simulated and observed water level at NOAA gauge 8546252.

R1C4:

4. The negatively correlated relationship between river discharge Q and ΔV (or Q and total water level TWL) in the Susquehanna River has to do with the fact that the imposed total water level was derived from the tidal gauge of 8545240, which is located in the Delaware River? Please clarify the adopted coastal boundary condition adopted in the Susquehanna River basin.

Author Response:

Our apologies for the confusion. For Susquehanna River, the total water level is obtained from the tidal gauge nearest to this river's outlet, i.e. gauge 8573364, located in Chesapeake Bay.

This is clarified in the introduction of the study domain (Line 119):

"the two tidal gauges near the river mouths, 8573364 and 8545240, provide the downstream boundary condition (BC) of the river model for Susquehanna River and Delaware River, respectively."

and section 5.1.1 (346-348):

"While ΔV is computed from the TWL simulation, Q and TWL are obtained from the paired streamflow and tidal gauges nearest to the river outlets, that is, gauge 01578310 and 8573364 for Susquehanna River and gauge 01463500 and 8545240 for Delaware River."

R1C5:

5. Since most cities are actually located in the downstream part of the adopted coastal boundary condition (e.g., 8545240 in Figure 1), the real application of flood control from this study focusing

on the river basin is questionable. Further explanation is required for the implication of this contribution.

Author Response:

We appreciate this comment.

One of the motivations to focus on the river basin is, as mentioned in Line 49-50, the interactions among backwater drivers and their respective contributions through fluvial processes are not well understood (Dykstra & Dzwonkowski, 2021). In the revised version, we added one more motivation:

"Although there are extensive literatures that address the storm surge induced coastal inundation (or flooding on land) and the related impacts on flood risks in coastal cities (Hinkel et al., 2014; Ye et al., 2020), limited efforts were made towards understanding the extreme surge that propagates into the river network (Ikeuchi et al., 2017). The latter is more critical in low-lying mega-delta regions that reside over 0.5 billion people globally (Syvitski & Saito, 2007)."

And in the discussion section (Section 6, Line 487), we elaborated the implication of this study:

"Thus, this study provides a basis for modeling coastal induced flooding in river basins. We aim to couple MOSART with the E3SM ocean model interactively in our next step for resolving the complex interactions at the river-ocean interface and eventually couple all land-river-ocean processes within E3SM to improve predictive understanding of compound flooding through pluvial, fluvial and coastal processes (Xu et al., 2022)."

Some minor comments:

R1C6:

1. Section 2.1: the study domain does not belong to Methodology.

Author Response:

We understand the reviewer's concern but prefer to keeping the description of study domain at this position as part of the methodology, as we think it might be more appropriate to provide readers a quick view of the study region followed by the regionally-refined mesh in section 2.3 and the occurred extreme events in section 2.7. In response, the title of Section 2 is changed to Methods and Data.

R1C7:

2. Lines 129-130: Please also illustrate the backwater effect in two studied regions.

Author Response:

Thanks. The documented backwater effects are added for Susquehanna River and Delaware river with references (Section 2.2, Line 139.

The reverse flow event in Mississippi River is used to demonstrate the extreme backwater scenario that the river flow direction reversed due to the elevated sea level.

"The extreme reverse flow is recently observed in Mississippi River during Hurricane Ida (Miller, 2021). In our study domain, the backwater processes in Susquehanna River and Delaware River were reported by USGS (U.S. Geological Survey, 2016) and showed significant impacts during Hurricane Irene (Zhang et al., 2020)."

R1C8:

3. Line 257: the definition of Kling-Gupta efficiency (KGE) is missing.

Author Response:

Thanks. The definition of KGE and the reference are now added to section 2.6.

The model performance is assessed using coefficient of determination ($r^2$), root mean square error (RMSE) and Kling–Gupta efficiency (KGE) (Gupta et al., 2009)

$$KGE = 1 - \sqrt{(r-1)^2 + \left(\frac{\delta_m}{\delta_o} - 1\right)^2 + \left(\frac{\sum X_m}{\sum X_o} - 1\right)^2}, \tag{9}$$

where $X_m$ and $X_o$ represents the model simulation and the observation, $\delta_m$ and $\delta_o$ are the corresponding standard deviations, and $r$ is their linear correlation.

R1C9:

4. Lines 282 and 288: eq. 1→Eq. 1

Author Response:

Corrected. (Line 300, 306)

R1C10:

5. Line 403: How did you compute the time series of $\Delta V$ (for the whole river basin or only in the main channel)?

Author Response:

We computed $\Delta V$ in the main channel because as mentioned in section 2.2 the routing of surface runoff in hillslopes and tributary still use kinematic wave method. In Equation 7 and 8, $h$ is specified as the main channel water depth, $L$ and $W$ are the length and width of the main channel.

We elaborated the definition of $\Delta V$ in section 2.6 (Line 240).

"The backwater effects are quantified by comparing the TWL and MSL simulations in terms of two quantification metrics along the main channel: water depth change $\Delta h$ and water volume change $\Delta V$:

$$\Delta h(t,i) = h_{exp}(t,i) - h_{MSL}(t,i),\tag{7}$$

$$\Delta V(t) = \sum_{i}^{N}\left(h_{exp}(t,i) - h_{MSL}(t,i)\right)L(i)W(i),\tag{8}$$

where the subscript $exp$ represents TWL, LFS or tide, $h$ is the main channel water depth, $t$ is the model output time step and $i$ is the grid cell index, $L$ and $W$ are the length and width of the main channel within the $i$th cell, respectively, and $N$ is the number of cells with nonzero $\Delta h$. The two metrics measure the backwater-induced changes within the river channel that are created from the variation in the downstream water level."

R1C11:

6. Figure 7: What are the observations for the grey solid and dashed line? In addition, please correct the units for the ΔV and the discharge

Author Response:

Figure 7 shows the extreme events overlaid on the corresponding time series. We rephrased the caption to increase the clarity and used dotted lines (instead of solid lines) to represent discharge. The units are also corrected.

[revised manuscript text omitted]

---

## Author Comment (AC2)

Response to Reviewers

Title: Investigating coastal backwater effects and flooding in the coastal zone using a global river transport model on an unstructured mesh

Author Response 1ˢᵗ revision

Reviewer 2
Reviewer Comments:

This paper presents a method to address the interactive phenomenon of river flooding and storm surge, using an unstructured mesh and focusing on two US coasts with different characteristics. The study underscores the role of backwater effects in representing the impact. Overall the study is well written and the results are clearly demonstrated. I have one minor comment.

Author Response:

We would like to sincerely thank the reviewer for the valuable comments and recommendations. We have carefully addressed the reviewer's suggestions as follows. The excerpts of the revised manuscript are provided as HESS does not allow us to share our revised revision at this stage of the review process.

R2C1:

P 5 L 135 Here the river widths are estimated using an empirical equation. This can be the source of uncertainty in simulations, so it should be relied only when other available and reliable data cannot be obtained. The target area is US, so there should be more reliable data. At least, global river width dataset has also been developed such as GWD-LR (Yamazaki et al. 2014). I would not ask the authors to re-calculate all the results, but just add some discussion on this point and consider using other data in future work.

Author Response:

We appreciate the reviewer comment and add the discussion of using more reliable river geometry dataset in future work:

"The river channel width and bankfull depth estimated from empirical formulations may introduce uncertainties. Even though such estimation achieves reasonable accuracy at local basins, more reliable river geometry data should be considered at least for regions wherever the data is available. While global river width datasets have been developed for river width >90 m (Allen & Pavelsky, 2018; Yamazaki et al., 2014), the river bankfull depth may also be derived from high-resolution remote sensing data. However, it remains challenging to upscale the observed river geometry to model resolution given the river is resolution free (Liao et al., 2022)."

Allen, G. H. and T. M. Pavelsky (2018). Global extent of rivers and streams. Science 361(6402): 585-588.

Liao, C., T. Zhou, D. Xu, R. Barnes, G. Bisht, H.-Y. Li, Z. Tan, T. Tesfa, Z. Duan and D. Engwirda (2022). Advances in hexagon mesh-based flow direction modeling. Advances in Water Resources 160: 104099.

Yamazaki, D., F. O'Loughlin, M. A. Trigg, Z. F. Miller, T. M. Pavelsky and P. D. Bates (2014). Development of the global width database for large rivers. Water Resources Research 50(4): 3467-3480.

---

## Author Response (AR1)

Response to Editor

Title: Investigating coastal backwater effects and flooding in the coastal zone using a global river transport model on an unstructured mesh

**Author Response 1st revision**

Editor Comments:

Dear authors,

I think you have adequately addressed the comments by the reviewers. You also presented revised paragraphs to be incorporated in the final paper. Provided you make these changes, i accept the revised paper.

**Author Response:**

Dear Editor,

Thank you for accepting our article. We have incorporated all the revisions (as specified in the response to reviewers) in the final paper. If you have any further questions or concerns, please let us know.

Regards,

Dongyu Feng

Pacific Northwest National Laboratory